# Isolable radical cation and dication of dialumene

Xufang Liu[1], Arseni Kostenko [1], Eva Körber [2], Huaiyuan Zhu[1], Karsten Meyer [2] & Shigeyoshi Inoue [1]✉

Alkenes are known to undergo successive oxidation to form alkene-derived radical cations and dications, which have found applications across various fields. As the aluminum analogues of alkenes, dialumenes likewise have the potential to lose one or two π-bonding electrons, forming dialumene-derived radical cations or dications. To date, however, these species have remained elusive, most likely due to the intrinsic electron deficiency imposed by both the positive charge and the pronounced electrophilicity of aluminum. Here, we present the synthesis of a stable aluminum-centered radical cation and dication through the combination of bulky silyl substituents and electron-donating carbene ligands. Further studies reveal that these aluminum complexes can switch between their neutral, radical cationic, and dicationic states, thus establishing a redox-reversible system. Furthermore, the dication exhibits multiple modes of reactivity, acting as a Lewis acid while also mediating both deoxygenation reactions and isocyanide homologation.

Alkenes ($R_2C=CR_2$), as the simplest π-conjugated systems, serve as electron reservoirs capable of undergoing sequential oxidation to generate alkene-derived radical cationic and dicationic species (Fig. 1a)[1–8], which significantly contribute to various fields including electrochemistry, catalysis, polymer chemistry, and materials science[9,10]. The isolation of these highly reactive compounds typically relies on stabilization provided by aromatic rings or heteroatom substituents. Over the past few decades, main-group multiple-bond chemistry has advanced rapidly, emerging as a powerful tool in modern synthetic chemistry[11–14]. Dialumenes (RAl=AlR), the aluminum analogues of alkenes, have shown great capabilities in molecular activation since their first isolation in 2017[15–23]. Removal of one or two π-bonding electrons from the aluminum centers should, in theory, afford dialumene-derived radical cations or dications.

In fact, aluminum-centered radicals are an exceptionally rare class of group 13 element-based radicals, in stark contrast to their lighter and heavier congeners, likely due to a confluence of aluminum's pronounced electrophilicity and the inherent reactivity of open-shell species[24–32]. Until now, only a few stable aluminum radicals have been isolated. Among these, aluminum radical anions **A**–**C** were

formed by populating the vacant p-orbital on aluminum (Fig. 1b)[33–36]. In addition, cAAC-stabilized aluminum species **D** and **E** are best described as aluminum-substituted neutral radicals with the unpaired electron mainly residing at the carbene carbon atoms, due to the capacity of cAAC for radical delocalization (Fig. 1c)[37–39]. Despite these advances, the isolation of aluminum radical cations is particularly challenging compared to their anionic and neutral counterparts, likely due to the increased electrophilicity stemming from their cationic nature.

Notably, low-valent cationic aluminum compounds remain rare, despite their potential ambiphilic reactivity[40–42]. Specifically, 1,2-dications containing an E–E single bond, are of utmost importance in organic synthesis[43]. Particularly, their Lewis acidity, combined with a filled E–E bonding orbital, underpins their rich and versatile chemistry. **F** can be defined as a cationic aluminum-centered cluster with aluminum atoms in the +I oxidation state, which forms a dimer in the solid state but exists as a monomer in dilute solution (Fig. 1d)[44]. However, aluminum-centered $Al^{II}$ 1,2-dications remain elusive, although their boron and gallium analogues have been reported[40,45–49], which might be also related to the intrinsic high electrophilicity of aluminum.

[1]Department of Chemistry, Institute of Silicon Chemistry and Catalysis Research Center, TUM School of Natural Sciences, Technische Universität München, Garching bei München, Germany. [2]Department of Chemistry and Pharmacy, Inorganic Chemistry, Friedrich-Alexander-Universität Erlangen-Nurnberg (FAU), Erlangen, Germany. ✉e-mail: s.inoue@tum.de

**Fig. 1 | Isolable Al-based radicals and low-valent Al cations. a** Radical cation and dication of alkene. **b** Examples of Al radical anions. **c** Examples of cAAC-stabilized Al-based neutral radicals. **d** Selected examples of low-valent Al cations. **e** This work: redox-reversible switching between dialumene, radical cation and dication.

We envisioned that the electron-rich nature and the flexible acyclic framework of base-stabilized dialumene would render it an ideal platform for accessing radical cation and dication through the respective one- and two-electron oxidations. Herein, we report the synthesis of stable aluminum-centered radical cation **2** and dication **3**, derived from a silyl-substituted dialumene **1** via a successive oxidation sequence (Fig. 1e). Conversely, dication **3** can undergo one-electron and two-electron reduction to regenerate the radical cation **2** and dialumene **1**, respectively. Meanwhile, the reaction of the dication **3** with dialumene **1** forms the radical cation **2** via comproportionation. Thus, the chemical behavior unveils a redox-reversible single-electron cycle between the three states for dinuclear group 13 complexes. Furthermore, we make use of the Lewis acidity and the often-overlooked low-valent character of the dication for reactivity studies, offering access to a broad spectrum of transformations.

## Results

### Synthesis and characterization

Building upon the successful isolation of the first neutral dialumene by our group in 2017[15,16], we targeted the use of SiTMS$_3$ substituent for the stabilization of a new dialumene. The central strategy is that the bulky SiTMS$_3$ substituent can provide both kinetic and thermodynamic stabilization through steric protection and the β-silicon effect, while the electron-rich carbene ligand can stabilize the electron-deficient cationic aluminum centers. Following this strategy, dialumene **1** was synthesized by the reduction of I$^i$Pr$_2$Al(SiTMS$_3$)I$_2$ with KC$_8$ and was isolated as a dark blue solid in 71% yield (Fig. 2a). Single-crystal X-ray diffraction (SC-XRD) analysis reveals that complex **1** adopts a trans-bent geometry (bent angle $\theta = 30°$) in the solid state (Fig. 2b, left, and Supplementary Fig. 47), which contrasts with the trans-planar geometry observed for previously reported Si$^t$Bu$_2$Me-dialumene[14]. In order to account for the differences between these two dialumenes, we performed quantum calculations using the ORCA software[50] (For details about the computational methods please refer to the Supplementary Information). We found that the Si$^t$Bu$_2$Me-dialumene can also exist in a trans-bent geometry similar to that of **1** with $\theta = 32.5°$ and

29.8° and an Al–Al bond length of 2.455 Å (Supplementary Fig. 59). The trans-bent geometry of Si$^t$Bu$_2$Me-dialumene is energetically almost identical to the trans-planar with a calculated $\Delta E$ of only 1.2 kcal mol$^{-1}$. So far, we could not obtain the trans-planar geometry of **1** as minimum. The frontier molecular orbitals of **1** are presented in Fig. 3. According to the NBO analysis of the canonical molecular orbitals, the HOMO-1 is a bonding orbital with the biggest contribution from the σ(Al–Al) and additional contributions from the σ(Al–Si) and σ(Si–Si) orbitals. The HOMO is also bonding and corresponds mainly to the π(Al–Al) bond, while the LUMO is the respective antibonding π*(Al–Al) orbital. The double bond character of the Al–Al interaction is manifested in the large Wiberg bond index of 1.54 (Fig. 2c). Full NBO analysis is presented in Supplementary Fig. 60.

The UV–Vis spectrum of **1** exhibits a strong absorption band in the visible region at 598 nm, which is slightly red-shifted relative to the Si$^t$Bu$_2$Me-dialumene ($\lambda = 573$ nm). TD-DFT (time-dependent density functional theory) calculations assign this band predominantly to HOMO → LUMO and HOMO → LUMO + 2 excitations (Supplementary Table 2).

To assess the feasibility of dialumene **1** to undergo oxidation, cyclic voltammetry (CV) was performed. Various electrolytes were screened, including NaBArF, [NBu$_4$][PF$_6$], NaBPh$_4$ and [NBu$_4$][BPh$_4$], of which [NBu$_4$][BPh$_4$] was found to be the most suitable (Supplementary Figs. 38–41). The resulting voltammogram displays two distinct redox events (Supplementary Figs. 40–44): the first is reversible (−2.0 V), and the second is irreversible (−1.4 V), suggesting the accessibility of the corresponding one-electron and two-electron oxidation products. These electrochemical results prompted us to examine the oxidation reactions of **1**. Reaction of **1** with one equivalent of the mild oxidant [Ph$_3$C][B(C$_6$F$_5$)$_4$] occurred readily in fluorobenzene at room temperature, and the radical cation **2** was isolated as a purple solid in 40% yield (Fig. 2a). The molecular structure of **2** reveals a slightly bent structure (bent angle $\theta = 6°$) and a highly twisted geometry around the Al–Al core (twist angle $\tau = 35.46°$) (Fig. 2b, middle, and Supplementary Fig. 47), markedly distinct from that of dialumene **1**. The Al–Al bond in **2** (2.4939(19) Å)

**a  Synthesis**

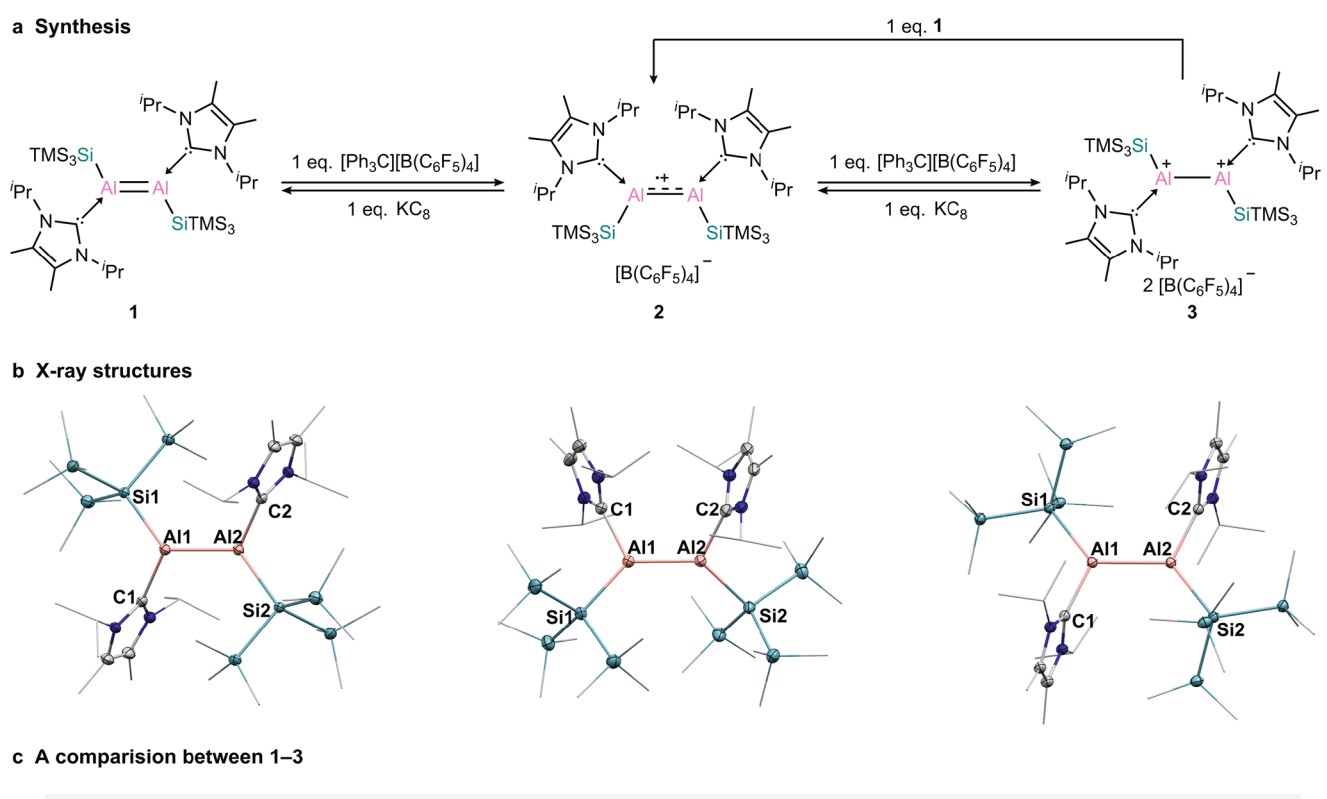

**b  X-ray structures**

**c  A comparision between 1–3**

| | Geometry | Al–Al [Å] | WBI (Al–Al) | Al–Si [Å] | Al–C [Å] |
|---|---|---|---|---|---|
| 1 | trans-bent ($\theta$ = 30°) | 2.463(4) | 1.54 | 2.504 (2) | 2.082(4) |
| 2 | bent and twisted ($\theta$ = 6°, $\tau$ = 35.46°) | 2.4939(19) | 1.29 | 2.4697(19), 2.4503(18) | 2.046(5), 2.056(5) |
| 3 | planar | 2.6612(10) | 0.87 | 2.4703(7) | 2.0352(18) |

**Fig. 2 | Synthesis of radical cation and dication and their redox chemistry. a** Synthesis of **2–3** and reversible interconversion between **1–3**. **b** X-ray structures of complexes **1–3** (anions for **2** and **3** were omitted for clarity). **c** Geometries, selected bond distances and Wiberg bond indexes (WBIs) of complexes **1–3**.

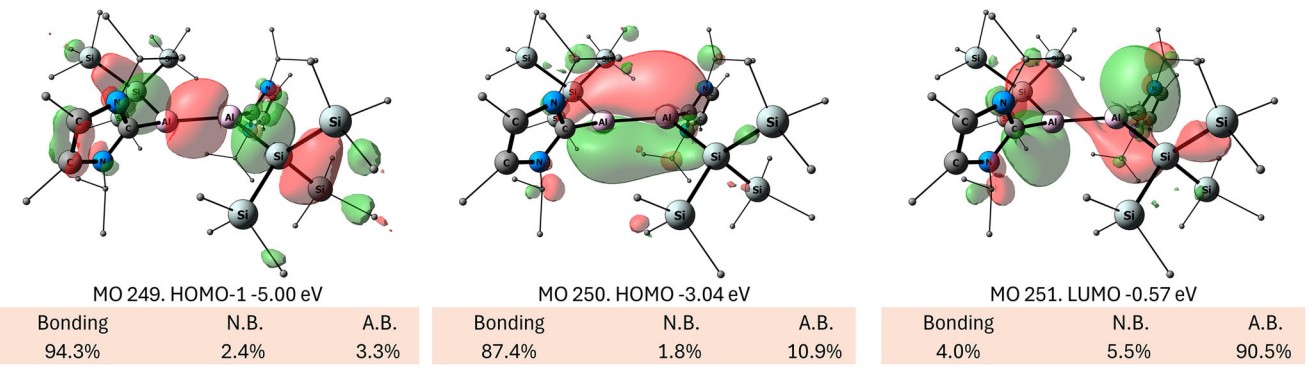

| MO 249. HOMO-1 -5.00 eV | | | MO 250. HOMO -3.04 eV | | | MO 251. LUMO -0.57 eV | | |
|---|---|---|---|---|---|---|---|---|
| Bonding | N.B. | A.B. | Bonding | N.B. | A.B. | Bonding | N.B. | A.B. |
| 94.3% | 2.4% | 3.3% | 87.4% | 1.8% | 10.9% | 4.0% | 5.5% | 90.5% |

**Fig. 3 | Frontier molecular orbitals of 1** (at the PBE0/def2-TZVP//B97-3c level of theory (iso = 0.03)).

is slightly elongated than that in **1** (2.463(4) Å), consistent with a formal bond order of 1.5.

The UV–Vis spectrum of **2** shows two intense absorption bands in the visible region. According to the TD-DFT calculations the band at 550 nm is attributed to αSOMO → αLUMO + 1 transition (Supplementary Table 3). The longest-wavelength absorption of **2** at 975 nm lies in the near-infrared region, significantly red-shifted compared to those of previously reported aluminum radicals[35,39]. This band corresponds to the αSOMO → αLUMO transition, which is red-shifted relative to the

corresponding HOMO-LUMO transition at 598 nm of neutral species **1**. The red shift may be attributed to the positive charge in **2**, which enhances Coulombic attraction in the excited state, where the electron in the αLUMO experiences a stronger interaction with the positively charged core, lowering the transition energy.

EPR analysis of **2** unambiguously confirms the presence of an unpaired electron at $g_{iso}$ = 1.9913 (Fig. 4a). The EPR spectrum displays well-resolved eleven signals due to the coupling of the unpaired electron with two magnetically equivalent $^{27}$Al nuclei ($I = ^5/_2$), which is in

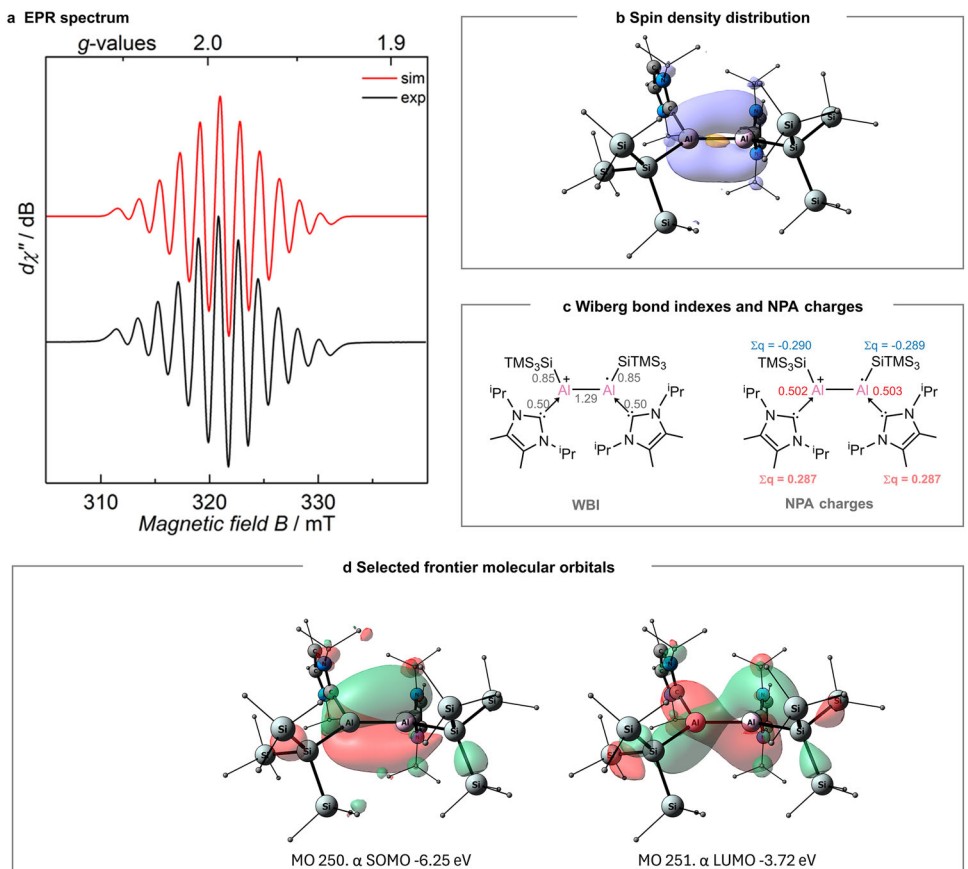

**Fig. 4 | EPR spectrum and computational analysis for complex 2. a** EPR spectrum of **2** recorded as a 3 mM ortho-DFB solution at room temperature (microwave frequency = 8.944 GHz). **b** Mulliken spin density surface (iso = 0.002). **c** Lewis structure with Wiberg bond indexes and Natural Population Analysis (NPA) charges. **d** Selected frontier molecular orbitals (at the PBE0/def2-TZVP//B97-3c level of theory (iso = 0.03)).

good accordance with the simulated spectrum (see more details in Supplementary Fig. 8). The hyperfine coupling constant ($A_{iso}$) was estimated to be 1.83 mT, which is larger than those of the radical anions **A** [$a(^{27}Al) = 1.19$ mT (R = CH(TMS)$_2$), $a(^{27}Al) = 1.04$ mT (R = Tipp)]. Although the s-character of SOMO is small, the large hfcc may also presumably be attributed to higher Coulombic attraction, which results in higher interaction between the unpaired electron and the nuclei. The Mulliken spin population of 0.53 on Al1 and 0.53 on Al2 indicate the formation of the Al–Al-centered radical. The spin density plot is presented in Fig. 4b. According to the EPR data and the calculations, **2** is best characterized as an aluminum-centered cationic radical species. The Al–Al moiety retains a double bond character with the WBI of 1.29, enabled by a single electron occupying the bonding π(Al–Al) orbital (Fig. 4c). The +1 positive charge is shared between the two Al centers having +0.50 el. each. The silyl substituents are slightly negatively charged with -0.29 el. each, while the carbene ligands are oxidized by 0.29 el. each. The reactivity of the radical cation **2** toward radical-trapping reagents, including TEMPO, $n$Bu$_3$SnH, HBpin and Ph$_2$Se$_2$, was further investigated. However, these reactions resulted in either no observable reactivity or produced ill-defined mixtures that could not be characterized.

Reaction of **1** with two equivalents of [Ph$_3$C][B(C$_6$F$_5$)$_4$] led to the clean formation of the dication **3** (Fig. 2a), which was isolated as a pale-yellow solid in 83% yield. SC-XRD analysis reveals that complex **3** adopts an almost perfect planar geometry, with a sum of angles around aluminum of 359.999 (Fig. 2b, right, and Supplementary Fig. 47). The Al–Al bond in **3** (2.6612(10) Å) is significantly elongated compared to that of **1**, falling within the typical range of Al–Al single bonds—a result

consistent with the removal of two electrons from the Al–Al π bond in **1**. The HOMO is a bonding orbital (Fig. 5, left), which corresponds the σ(Al–Al) as well as σ(Al–Si) and σ(Si–Si) orbitals (see also Supplementary Fig. 62). The LUMO is non-bonding composed to the largest extent of the empty p orbitals of the Al centers. In the dication, the Al–Al bond order further decreased, in comparison to **1** and **2**, with WBI of 0.87, indicating the single bond character (Fig. 5, right). The positive charge is mostly located on the two aluminum atoms. The carbene ligands, which donate electrons to stabilize the Al centers are oxidized by +0.42 el., while the silyl substituents are slightly negatively charged with −0.24 el., withdrawing electrons from the Al centers. NBO analysis shows rather high occupancy of the lone vacancy orbitals at the Al centers of 0.15 el. According the second order perturbation theory analysis, the stabilization of the vacant p orbitals of the Al centers is achieved not via the hyperconjugation of σ(Si–Si) as we initially intended – this effect is very small in the dicationic complex -3 kcal mol$^{-1}$ for each Al center—but rather by substantial agostic-like interactions between the empty p-orbital of Al and the methyl C-H bonds of the NHC isopropyl groups. These sum up to -40 kcal mol$^{-1}$ for each Al center (Supplementary Fig. 63).

## Redox-reversible interconversions

Redox-reversible processes are the hallmark of transition metal catalysis, yet such processes remain rare among main-group compounds[51–61]. Having established that dialumene **1** can undergo successive oxidation to form the mixed-valence radical cation **2** and the doubly-oxidized dication **3**, we turned to investigate the reverse reduction processes using KC$_8$ as the reductant (Fig. 2a). Reaction of

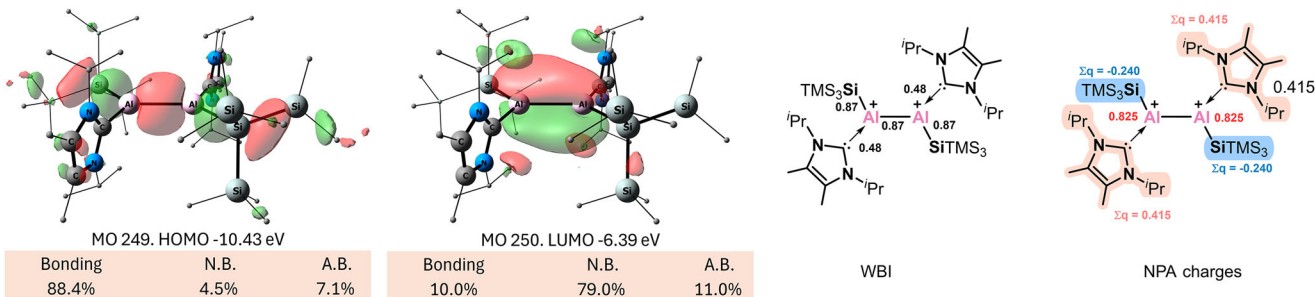

**Fig. 5 | Computational analysis for complex 3.** Left - frontier orbitals of **3** (at the PBE0/def2-TZVP//B97-3c level of theory (iso = 0.03)); right - WBI and NPA charges of **3** (at the PBE0/def2-TZVP//B97-3c level of theory).

the dication **3** with KC$_8$ resulted in a sequential formation of the one-electron reduced radical cation **2** and the two-electron reduced dialumene **1**. In addition, the reaction of the dication **3** with dialumene **1** proceeded through comproportionation to generate the radical cation **2**, in which **1** is formally oxidized while **3** is reduced. Consequently, **1** together with **2** and **3** forms an aluminum-centered redox system which can reversibly shift between its neutral, radical cationic and dicationic states. Such an isolable redox-reversible system, capable of cycling in triple oxidation states, has not yet been established for dinuclear group 13 complexes (E13–E13). These results undoubtedly demonstrate the ability of complexes **1**–**3** to enable electron-transfer reactions that are oftentimes invoked in organometallic chemistry and catalysis[62].

**Reactivity studies of the dication**

The intrinsic Lewis acidity and the low-valent character of the dinuclear dications motivated us to investigate their reactivities. The general concept is that the empty p orbital of the Al$^{II}$ atom can accept electrons from a Lewis base, while the two adjacent low-valent Al$^{II}$ centers, upon Al–Al bond cleavage, can facilitate substrate insertion. Despite this, research on cationic compounds has primarily centered on their Lewis acidity, which limited their widespread applications in small-molecule activation[44,46].

We began our studies by exploring the reaction of **3** with Lewis bases, aiming to uncover the high Lewis acidity of the Al$^{II}$ dication (Fig. 6). Reaction with DMAP yielded the base-stabilized dicationic product **4a**, featuring aluminum atoms in the +II oxidation state. Each aluminum atom is tetracoordinate, thus complex **4a** can be termed as a dialuminum dication possessing an Al(sp$^3$)–Al(sp$^3$) bond. In the solid state, the Al–Al bond distance is 2.7596(10) Å (Fig. 7), which is considerably longer than those in three-coordinate dialuminum compounds, due to the increased steric congestion around the Al–Al core in **4a**. Similarly, reaction with pyridine yielded the corresponding Al$^{II}$ dicationic product **4b**. In the solid state, the Al–Al bond distance is 2.777 Å (Fig. 7), comparable to that in **4a**. Additionally, reaction of **3** with benzonitrile led to clean generation of the Lewis adduct **5**, with an elongated Al–Al bond of 2.701 Å. These reactivities certainly confirm that the central aluminum atoms in the dication **3** act as electrophilic sites for complexation with Lewis bases.

Next, we set out to explore the potential of dication **3** for substrate activation, beyond classical Lewis acid-type reactivity, harnessing its low-valent nature and the cooperative interplay between its two electrophilic aluminum centers. The Al$^{II}$ dication is expected to be formally oxidized to Al$^{III}$ dication upon coupling with substrates, accompanied by cleavage of the Al–Al bond.

As anticipated, treatment of **3** with N$_2$O at room temperature overnight afforded the oxygen-bridged Al$^{III}$ dication **6**. SC-XRD analysis reveals a central linear Al–O–Al moiety (Fig. 7), in which the Al–O bond distance of 1.677 Å compares well to those in ligated dialuminoxanes[63,64]. Motivated by this finding, we then investigated its

oxygen-abstracting ability using other oxygen sources. Reaction with pyridine-N-oxide immediately formed the pyridinium-ligated dication **7**, with the positive charges located on the nitrogen atoms. SC-XRD analysis reveals a bent Al–O–Al moiety (Al1–O1–Al2: 154.11°), with an average Al–O bond distance of 1.716(8) Å (Fig. 7), close to that in complex **6**. Notably, compound **7** forms much faster than **6**, suggesting that the reaction first generates a highly reactive double pyridine-N–O-coordinated intermediate, which subsequently abstracts an additional oxygen from pyridine-N-oxide to yield **7**. Complex **7** is unstable in solution and gradually converted to the pyridine-coordinated dication **8** via oxygen migration into the Al–Si bonds. Complex **8** features a central linear Al–O–Al moiety, with Al–O bond distance of 1.6904(7) Å (Fig. 7), very close to that in complex **6**. The core structure in **7** and **8** can be considered as a [Al$_2$O$_3$] chain, reflecting the strong oxophilicity of the dication **3**. Particularly, the Si–O–Al–O–Al–O–Si linkage in complex **8** models a single repeating unit of the zeolite framework.

We then examined the reactivity of **3** toward carbon monoxide, but it yielded an unidentified mixture. Isocyanides are isoelectronic analogues to carbon monoxide, and like CO are versatile C1 building blocks in organic synthesis. The activation of isocyanides by main group elements has emerged as a fascinating field in modern synthetic chemistry[65]. However, the use of cationic compounds in this context remains largely unexplored. Reaction of **3** with isocyanides, such as XylNC and MesNC, afforded the Al$^{III}$ dications **9a** and **9b**, respectively. The molecular structures of **9a** and **9b** were determined by SC-XRD analysis (Fig. 7), in which an N=C–C=N unit bridges the two separated Al$^{III}$ fragments, resembling the product of a neutral dialane reacting with isocyanide[66,67]. To the best of our knowledge, this represents a rare case of isocyanide homologation mediated by a cationic aluminum species[65], in sharp contrast to the previously reported diboron dication that results in a monomeric boronium species[47]. The isolation of compounds **6**–**9** may arise from the intrinsic Lewis acidity of the dication and the interaction between its two cationic centers.

For comparison, the reactivity of radical cation **2** toward N$_2$O was examined, affording an oxygen-bridged dication **6** (Supplementary Fig. 46), likely via an aluminum-mediated electron-transfer pathway.

Taken together, these results demonstrate that the dication **3** can behave as a potent Lewis acid while also mediating deoxygenation and isocyanide homologation reactions. These reactivities showcase the capacity of Al$^{II}$ dication for substrate coordination and subsequent insertion, mirroring the typical profiles of transition metals and highlighting its potential as a sustainable substitute for transition-metal complexes in small-molecule activation and catalysis[13].

In summary, this study describes the synthesis and isolation of a stable aluminum-centered radical cation and dication through successive oxidation of dialumene bearing a bulky silyl substituent. DFT studies reveal that agostic-like interactions between the vacant p orbitals of aluminum centers and the methyl C-H bonds of the NHC isopropyl groups play a crucial role in stabilizing the dicationic species. Further studies show that these aluminum complexes can participate

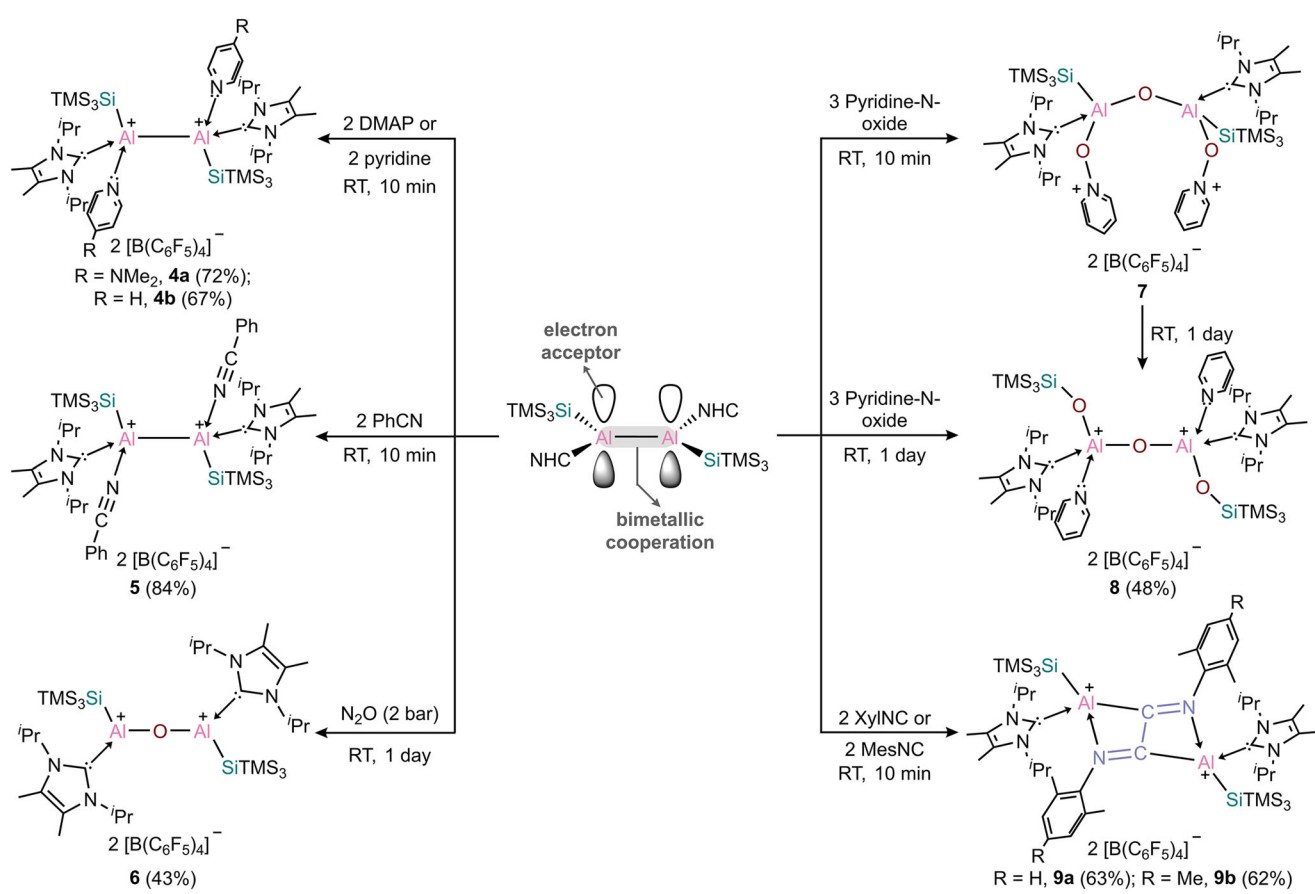

**Fig. 6 | Reactivity studies of dication 3.** Multiple reactivities of **3** toward Lewis bases, oxygen sources and isocyanides.

in a redox-reversible one-electron shuttling process, which interconverts in neutral, radical cationic and dicationic states. These findings contribute to a deeper understanding of metal complexes-mediated electron-transfer reactions, a topic of critical importance for dictating reactivity and controlling catalytic turnover. Furthermore, the dication exhibits diverse reactivity, which can not only bind Lewis bases but also facilitate substrate insertion, leveraging its intrinsic electrophilicity and low-valent character. This transition-metal-like reactivity pattern demonstrates the viability of cationic aluminum complexes as platforms for catalytic transformations.

## Methods
### Synthetic methods
All experiments and manipulations were carried out under an argon atmosphere using standard Schlenk or glovebox techniques. The glassware was heat-dried under vacuum prior to use. Solvents were dried by standard methods (withdrawal from MBraun Solvent Purification System and storage over molecular sieves, or distilled from sodium/benzophenone or $CaH_2$ under argon atmosphere and degassed via freeze-pump-thaw cycling). Standard chemicals were purchased from commercial suppliers and used as received if not stated otherwise.

### Spectroscopic methods
All NMR samples were prepared under argon in J. Young PTFE tubes. NMR spectra were recorded on a Bruker AV400US, DRX400, AVHD300 and AV500cr at ambient temperature if not stated otherwise. $^1H$ and $^{13}C$ NMR spectra were calibrated against the residual proton and natural abundance carbon resonances of the respective deuterated solvent as internal standard. Quantitative elemental analyses (EA) were carried out using a EURO EA (HEKA tech) instrument

equipped with a CHNS combustion analyzer at the Laboratory for Microanalysis at the TUM Catalysis Research Center. The UV-vis spectra were taken on an Agilent Cary 50 spectrophotometer with a Schlenk quartz cuvette at the Central Analytic Department at the TUM Catalysis Research Center. EPR spectra were recorded on a JEOL continuous-wave spectrometer JES-FA200, equipped with an X-band Gunn diode oscillator bridge, a cylindric mode cavity, and a helium cryostat. Electrochemical measurements were carried out at room temperature under dinitrogen atmosphere with an μAutolab Type III potentiostat.

### Crystallographic methods
Single crystal diffraction data were collected on a single-crystal X-ray diffractometer equipped with a Charge-Integrating Pixel Array Detector (Brucker Photon-II), a Microfocus X-Ray Source with a $CuK_\alpha$ ($\lambda = 1.54178$) or a Turbo X-Ray Source rotating anode with $MoK_\alpha$ radiation ($\lambda = 0.71073$ Å) and a Helios optic using the APEX4 software package. The measurements were performed on single crystals coated with the perfluorinated ether Fomblin Y. The crystals were fixed on the top of a micro sampler, transferred to the diffractometer and frozen under a stream of cold nitrogen. Additional details for data processing, structure refinement and graphic depictions are given in the Supplementary Information.

### Computational methods
Calculations were carried out using ORCA 6 software[50]. Geometry optimizations were carried using the B97-c composite method. The optimized geometries were verified as minima by analytical frequency calculations. The NBO analysis was done using the NBO7 software, at the PBE0/def2-TZVP//B97-c level of theory. TD-DFT calculations were carried out at the B3LYP/6-311G(2d,2p)/SMD=THF//B97-c level of theory.

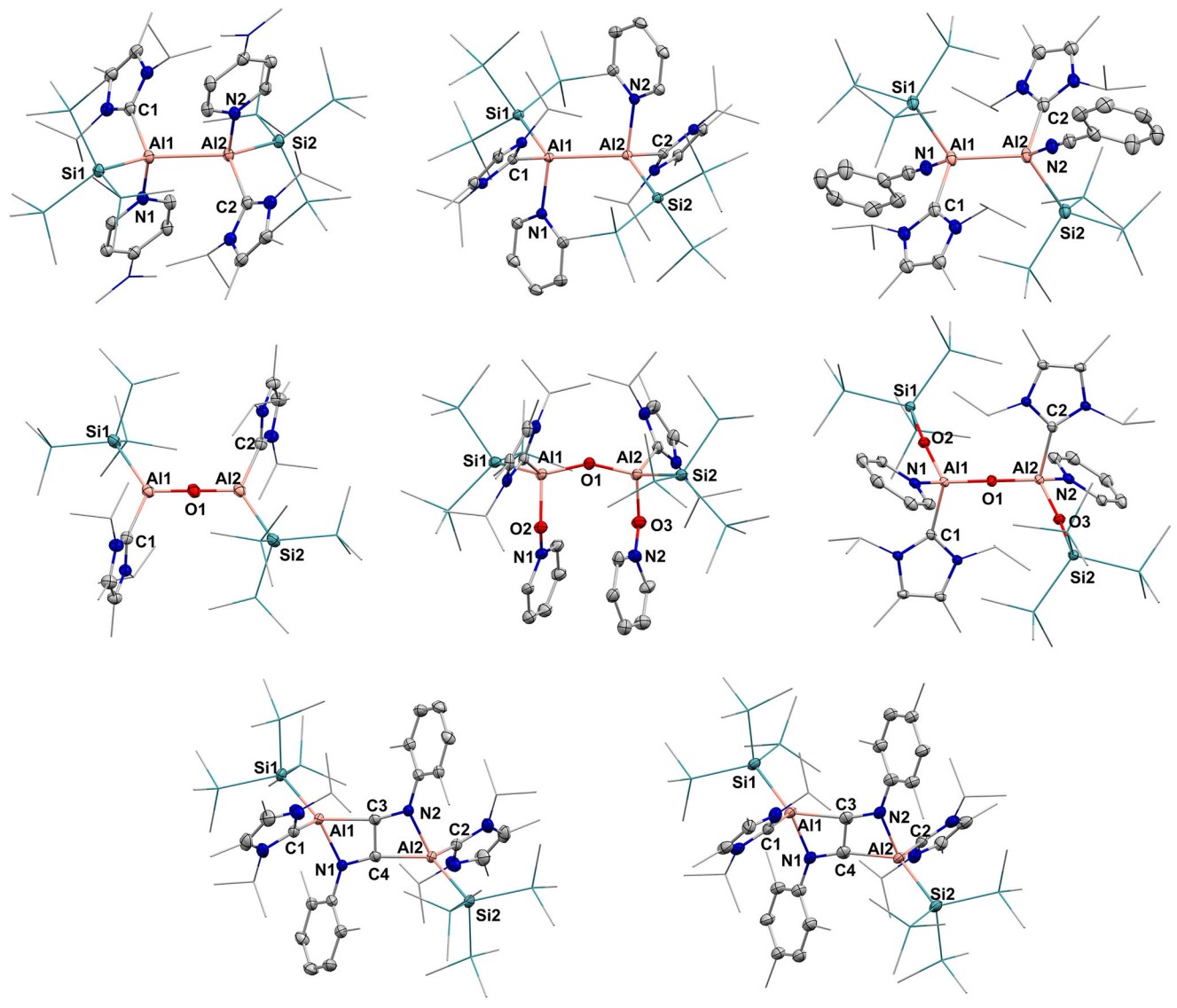

**Fig. 7 | X-ray structures of complexes 4–9.** Anions were omitted for clarity.

## Data availability

The data that support the findings of this study are available within the main text and its Supplementary Information. Crystallographic data for the structures reported in this Article have been deposited at the Cambridge Crystallographic Data Centre, under deposition numbers CCDC 2490063 (**1**), 2490064 (**2**), 2490065 (**3**), 2490066 (**4a**), 2490067 (**4b**), 2490068 (**5**), 2490069 (**6**), 2490070 (**7**), 2490071 (**8**), 2490072 (**9a**), and 2490073 (**9b**). Copies of the data can be obtained free of charge via https://www.ccdc.cam.ac.uk/structures/. All data are available from the corresponding author upon request. Source Data are provided with this manuscript. Source data are provided with this paper.

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

## Acknowledgements

This project has received funding from the Alexander von Humboldt foundation for Research Fellowships (to X.L.), as well as the European Union's Horizon 2020 research (ALLOWE101001591) and innovation programme under the Marie Skłodowska-Curie grant agreement No 899987. The authors gratefully acknowledge the scientific support and HPC resources provided by the Erlangen National High Performance Computing Center (NHR@FAU) of the Friedrich-Alexander-Universität Erlangen-Nürnberg (FAU) under the NHR project b255bb. NHR funding is provided by federal and Bavarian state authorities. NHR@FAU hardware is partially funded by the German Research Foundation (DFG) – 440719683. We thank Friedrich-Alexander-Universität Erlangen-Nürnberg (FAU) for generous funding.

## Author contributions

X.L. and H.Z conceived and performed the synthetic experiments and analysed the data. A.K. designed and performed the theoretical analyses. E.K. and K.M. conceived and performed the CV and EPR measurements. S.I. conceived and supervised the project. X.L., A.K., and S.I. wrote the manuscript with input and critical revision from all authors.

## Funding

## Competing interests

The authors declare no competing interests.
