## [Transparent Peer Review file · Nature Communications]

Isolable Radical Cation and Dication of Dialumene

Corresponding Author: Professor Shigeyoshi Inoue

Version 0:

Reviewer comments:

Reviewer #1

(Remarks to the Author)

The authors have provided a very detailed response to the original reviews which is appreciated. However, much of this information hasn't necessarily translated into the manuscript. This specifically relates to much of the reactivity studies discussed in their response and the electrochemistry data. These reactions and experimental detail are of interest to the community, and the manuscript would be strengthened for the inclusion. I will trust the authors to make a judgement here as to what information to include in the manuscript versus the SI – but I would encourage them to use the supporting information as a space to include more detailed discussion relating to the compounds presented, but which is perhaps beyond the narrative of the main text.

Overall, I feel the manuscript is of broad interest to main group community, the work has been conducted to a high standard, and the molecules certainly add another dimension to the rich chemistry of group 13 elements which is currently being discovered. Once the above points are addressed, and I support publication in Nature Communications.

Reviewer #2

(Remarks to the Author)

The manuscript by Inoue and co-workers presents the first examples of stable aluminum-centered radical cation and dication species, achieved through the combined use of a bulky silyl substituent and an electron-rich carbene ligand. These complexes undergo reversible interconversion between neutral, cationic, and dicationic states, thereby establishing a rare and instructive case of a redox-switchable main-group system. The dication exhibits diverse reactivity, functioning as a Lewis acid and a mediator in isocyanide homologation, thus broadening the redox chemistry accessible to low-valent aluminum compounds.

In my view, this is (still) a high-quality and carefully executed study. The revised version addresses several issues raised during the initial review. In particular, the problem with the EPR spectrum has been resolved, and additional cyclic voltammetry measurements have been carried out. The experimental work is thorough, the conclusions are well supported by computational analysis, and the overall presentation is clear. I therefore consider the manuscript suitable for publication. However, it is ultimately for the editorial office to decide whether the concerns raised by the other referees in the original submission have been addressed satisfactorily.

Reviewer #3

(Remarks to the Author)

The authors appropriately revised the manuscript according to the results of the first round of review. Therefore, the reviewer suggests accepting this paper as it stands.

REVIEWERS' COMMENTS

Reviewer #1 (Remarks to the Author):

The authors have provided a very detailed response to the original reviews which is appreciated. However, much of this information hasn't necessarily translated into the manuscript. This specifically relates to much of the reactivity studies discussed in their response and the electrochemistry data. These reactions and experimental detail are of interest to the community, and the manuscript would be strengthened for the inclusion. I will trust the authors to make a judgement here as to what information to include in the manuscript versus the SI – but I would encourage them to use the supporting information as a space to include more detailed discussion relating to the compounds presented, but which is perhaps beyond the narrative of the main text.

Overall, I feel the manuscript is of broad interest to main group community, the work has been conducted to a high standard, and the molecules certainly add another dimension to the rich chemistry of group 13 elements which is currently being discovered. Once the above points are addressed, and I support publication in Nature Communications.

Response: We thank the reviewer for the valuable suggestions. As recommended, we have added a brief discussion of the reactivity of the radical cation toward N_2O in the revised manuscript. Regarding the electrochemical data, we now state in the revised main text that several electrolytes were screened, including NaBArF, $[NBu_4][PF_6]$, NaBPh₄, and $[NBu_4][BPh_4]$, of which $[NBu_4][BPh_4]$ was found to be the most suitable. The corresponding cyclic voltammograms have been included in the revised Supporting Information.

Reviewer #2 (Remarks to the Author):

The manuscript by Inoue and co-workers presents the first examples of stable aluminum-centered radical cation and dication species, achieved through the combined use of a bulky silyl substituent and an electron-rich carbene ligand. These complexes undergo reversible interconversion between neutral, cationic, and dicationic states, thereby establishing a rare and instructive case of a redox-switchable main-group system. The dication exhibits diverse reactivity, functioning as a Lewis acid and a mediator in

isocyanide homologation, thus broadening the redox chemistry accessible to low-valent aluminum compounds.

In my view, this is (still) a high-quality and carefully executed study. The revised version addresses several issues raised during the initial review. In particular, the problem with the EPR spectrum has been resolved, and additional cyclic voltammetry measurements have been carried out. The experimental work is thorough, the conclusions are well supported by computational analysis, and the overall presentation is clear. I therefore consider the manuscript suitable for publication. However, it is ultimately for the editorial office to decide whether the concerns raised by the other referees in the original submission have been addressed satisfactorily.

Response: We are grateful to the reviewer for the positive feedback.

Reviewer #3 (Remarks to the Author):

The authors appropriately revised the manuscript according to the results of the first round of review. Therefore, the reviewer suggests accepting this paper as it stands.

Response: We sincerely thank this reviewer for the recommendation.